# Estimation of the Equivalent Circuit Parameters of Induction Motors by Laboratory Test

**Moshe Averbukh *** **and Efim Lockshin**

Department of Electrical and Electronic Engineering, Ariel University, Ariel 44837, Israel; yafimlo@ariel.ac.il
\* Correspondence: mosheav@ariel.ac.il; Tel.: +972-528-814-120

**Abstract:** The determination of equivalent circuit parameters for AC induction motors represents an important task in an electrical machine laboratory. Frequently used open-circuit and short current tests answer these requirements. However, the results have a low accuracy. This becomes especially obvious when the equivalent circuit is applied for the motor current and power prediction. The main obstacles in this circumstance lie in the difficulty of providing a pristine open-circuit test, the lack of which causes errors in parameter estimation. A much more accurate approach can be carried out with a test including several output points with measurements of the motor torque, velocity, current, and power magnitudes. Nevertheless, a relatively simple and accurate method to ensure determining parameters for such tests does not exist. This article tries to provide such a method by an approach based on Kloss's simplified equation and the Thevenin theorem. The significant novelty of the method is the specially selected synergetic interaction between the analytical and numerical approaches, which give a relatively simple algorithm with a good accuracy and a convergence of the parameters' estimation.

**Keywords:** AC induction motor; equivalent circuit parameters estimation; Thevenin theorem

## 1. Introduction

AC induction motors, owing to their compactness, high reliability, and low costs, are found in a large number of industrial machines. The use of these motors continues to become increasingly widespread due to the availability and decreasing cost of variable frequency drives (VFD), emphasizing the superiority of asynchronous electrical motors over other types. Therefore, the requirements for selecting optimal AC motors need greater attention. The optimal choice of AC motor can be effectively carried out by the equivalent circuit that can ensure estimation of motor output characteristics in a wide range of conditions. The determination of AC induction motor equivalent circuit parameters represents one of the important laboratory tasks both in the study of electrical machines in academic courses of electrical machines and drives [1], and industrial applications [2–4].

There are three major groups of methods for the estimation of AC induction motor parameters [5–11]. The first one includes software programs such as ETAP [5], numerous methods based on a simplified circuit algorithm [6], nameplate data [7], and manufacturers' datasheets [8–11]. The application of these methods, both for students and technicians in a production laboratory, is problematic. This is due to difficulties and complications in the use of special software and the high cost of these programs. An additional obstacle to using special software and methods in the educational laboratory, described in [5–11], is the advanced age of the electrical machines used in such study and the fact that they have no fully rated data.

The second group of methods of assessing equivalent AC motor parameters is represented by different experimental techniques [12–18]. Wang et al. [12] suggested a nonlinear least-squares approach for the identification of equivalent circuit parameters. Yamamoto et al. [13] used no-load and locked-rotor tests with a simple procedure to determine equivalent circuit parameters

of an induction double-cage rotor motor. An estimation of equivalent circuit parameters of an induction motor from load data was proposed by Bhowmick et al. [14]. De Silva used linear graphs and the Thevenin–Norton circuits theorem for an analysis of electro-mechanical systems including induction motors [15]. Daut et al. [16] assessed specific 0.5 HP motor equivalent parameters based on a load factor test. The searching procedure established on stochastic principles of particle swarm optimization for finding AC motor equivalent circuits was proposed by Sakthivel et al. in [17] and by Nikranajbar et al. in [18].

It is worth mentioning the analytical formulations and methods for AC motor parameter identifications. Among them is the Kloss formula, which simplifies the definition of the torque vs motor velocity slip [19–21]. Panovko and Shokhin used Kloss expression for the analysis of the AC induction motor in a vibrating machine [19]. Mihăilescu and Galan applied the Kloss rule for the assessment of output parameters and performances of a three-phase induction motor fed by a vector control [20]. Andersson et al. used Kloss formulation for selecting the appropriate AC motor in the design of aircraft actuation systems [21].

The third group of approaches to determining the parameters and functionality of an AC asynchronous motor uses the application of the Thevenin theorem [22–24]. Haque used the Thevenin circuit for the determination of the parameters of NEMA induction motors with manufacturer datasheets [22]. Abidin and Adam employed a Thevenin diagram for the design of the AC motor prototype [23]. Mohammadi and Akhavan used a three-phase motor hybrid approach combining the Thevenin theorem with a genetic algorithm and particle swarm optimization for a parameter estimation [24].

The literature review provides several effective and rather accurate examples of the estimation of AC motor equivalent circuit parameters based on dissimilar methods. However, for the education or production laboratory in the manufacturing facilities of electrical machines, they are not exactly applicable, as they are either unnecessarily complex or expensive. The method for parameter finding should be rather simple but sufficiently accurate in providing the results of AC motor testing.

This article proposes an original approach based on the data of motor loading, the Kloss equation, and the Thevenin circuit. This method provides a simplified technique for experimental testing that ensures accurate results for those who carry out this work. The article includes the following sections. A methodology section represents the definition of output parameters of induction motors by a T-equivalent circuit, Thevenin simplification of the equivalent circuit, and Kloss-formulation-related maximum achievable motor torque, as well as a critical slip with results of motor loading. In addition, the methodology section provides the foundation of an equivalent circuit estimation based on the synergetic cooperation of the developed analytical and numerical methods. Furthermore, the methodology section describes the definition of two important motor parameters—$\tau_{max}$ and $s_K$—based on measured torque–velocity points during motor loading. The results section shows the experimental laboratory setup that was used for the verification of the proposed approach, as well as the confidence of this method's application based on the comparison of the theoretical output motor performance and motor current with measured parameters. The last sections summarize the results of the work that was done in the study.

## 2. Methodology

### 2.1. AC Induction Motor and Output Parameters Defined by an Equivalent T-Circuit

The functionality of AC induction motors is described in different ways. The dynamic formulation of the electromechanical processes during electromagnetic interactions between motor windings was found by Kraus et al. [25] and Leonhard [26]. This methodology ensures the most accurate results of dynamic processes. However, its application is problematic for students and technicians in manufacturers' facilities, as it requires a profound knowledge of mathematics and high technical ability to operate complex software programs.

Another way of describing an AC machine, known as the energy analysis approach, is based on the analysis of the energy flows diagram of an electric machine. This method,

described by Chapman [27], gives satisfactory and quite precise results for the steady-state mode of electrical machines' functionality. The major representative of the energy approach is an equivalent motor circuit, different variations of which were presented by Toliyat and Kliman [28]. Among them, the most useful is the T-circuit (Figure 1). We adopted this diagram for our investigation because it can be used for the verification of all AC input and output characteristics in a steady-state mode.

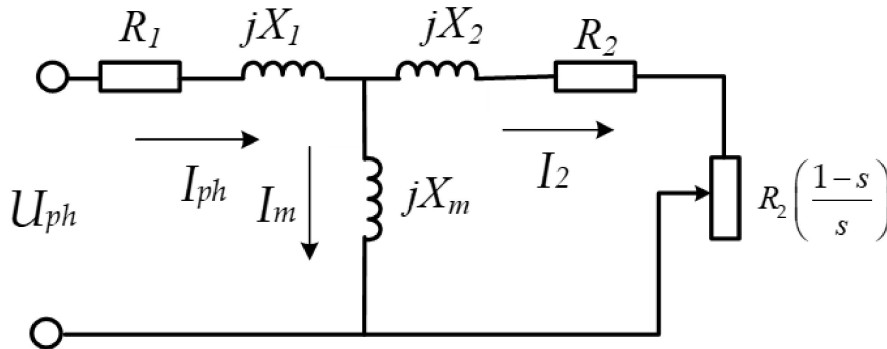

**Figure 1.** T-type equivalent circuit of an AC induction motor.

The equivalent circuit is related to one phase of a motor and includes connections between $R_1$ and $R_2$; the equivalent resistances of a stator and rotor circuits, $X_1$, $X_2$; the reactance of leakage flux in stator and rotor magnetic circuits, and $X_m$; the reactance of the magnetizing flux. The equivalent circuit allows the calculation of the main motor input–output parameters—such as the input current, power, power factor (*cos φ*), output power, and output characteristic—as the function of rotation the velocity vs. motor torque as follows. The magnitudes of the input parameters are:

$$(I_{in})_{ph} = \frac{U_{ph}}{Z_{in}}; \quad Z_{in} = R_1 + jX_1 + \frac{jX_m\left(\frac{R_2}{s}+jX_2\right)}{\frac{R_2}{s}+j(X_m+X_2)} = R_1 + jX_1 + \frac{jX_m(R_2+jX_2s)}{R_2+js(X_m+X_2)}$$

$$P_{in} = 3(I_{in})_{ph} \cdot U_{ph} \cdot cos\varphi$$

$$cos\varphi = cos(arg(Z_{in}))$$

(1)

The estimation of the output parameters is rather complicated with the T-circuit having two serial–parallel branches. Nevertheless, the complexity of the calculations can be eliminated by applying the Thevenin approach. As a result, a simpler Thevenin-equivalent circuit of the AC motor can be obtained, as shown in Figure 2.

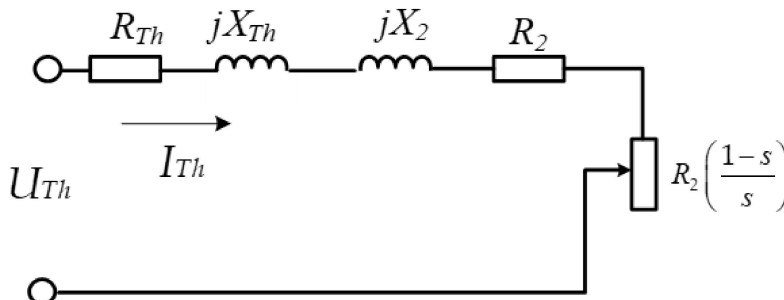

**Figure 2.** Thevenin one-phase equivalent circuit for an AC motor.

The Thevenin voltage, resistance, and reactance are calculated as per Chapman [27]:

$$U_{Th} = \frac{X_m}{X_m + X_1} = \gamma;$$

(2)

$$R_{Th} = R_1 \left( \frac{X_m}{X_m + X_1} \right)^2 = R_1 \cdot \gamma^2; \tag{3}$$

$$X_{Th} = X_1 \tag{4}$$

Thevenin's approach, along with the motor output characteristic, allows the determination of the maximum torque $\tau_{max}$ (5) and a critical slip $s_K$ (6):

$$\tau_{max} = \frac{\frac{3}{2} \frac{U_{Th}^2}{\omega_S}}{R_{Th} + \sqrt{R_{Th}^2 + (X_{Th} + X_2)^2}}; \tag{5}$$

$$s_K = \frac{R_2}{\sqrt{R_{Th}^2 + (X_{Th} + X_2)^2}}; \tag{6}$$

It is worth pointing out that those Equations (5) and (6) form an algebraic system of equations, the solution to which provides a basis for finding the parameters $R_{Th}$, $R_2$, $X_{Th}$, and $X_2$. However, these equations can be solved only if the $\tau_{max}$ and $s_K$ are known. To find $\tau_{max}$ and $s_K$, we used the Kloss expression relating the motor torque and motor velocity so:

$$\frac{\tau_m}{\tau_{max}} = \frac{2}{\frac{s}{s_K} + \frac{s_K}{s}} \tag{7}$$

The Kloss formulation can be applied to the assessment of $\tau_{max}$ and $s_K$ by fitting them with an LMS procedure to a set of operating points obtained from experimental measurements.

The precision of the motor output characteristic by a Kloss formula is relatively high (the discrepancy between theoretical and real characteristics is less than 1–2%) in the range where torque is changing from zero to a maximum value. The accuracy of the motor description by a Kloss expression is lower when the slip $s$ is changing from $s_K$ toward $s_{max}$ ($s_{max} = 1$). However, this does not lessen the accuracy of the assessment of $\tau_{max}$ and $s_K$, as the fitting is carried out during the first interval of the motor characteristic beginning at zero and increasing to maximum torque.

To facilitate the solution of (5) and (6), the parameters $R_{Th}$, $R_2$, $X_{Th}$, and $X_2$ were combined into dimensionless variables $\alpha$ and $\beta$ as follows:

$$\tau_{max} = \frac{\frac{3}{2} \frac{U_{Th}^2}{\omega_S R_2}}{\frac{R_{Th}}{R_2} + \sqrt{\left( \frac{R_{Th}}{R_2} \right)^2 + \left( \frac{X_{Th} + X_2}{R_2} \right)^2}} = \frac{\frac{3}{2} \frac{U_{Th}^2}{\omega_S R_2}}{\alpha + \sqrt{\alpha^2 + \beta^2}}; \tag{8}$$

$$s_K = \frac{1}{\sqrt{\alpha^2 + \beta^2}} \tag{9}$$

The use of dimensionless parameters $\alpha$ and $\beta$ allows one to obtain their values by solving the algebraic system (8) and (9). The knowledge of $\alpha$ and $\beta$ provides the foundation of the method of estimating parameters that is represented below.

### 2.2. Foundation

The proposed method uses a set of operating points obtained through a motor loading diagram. For the demonstration of the method, the set of output points of the loading diagram are represented in Table 1. The submitted method is explained by the algorithm in Figure 3. In the first step, the measured points of motor loading are approximated using the Kloss Formula (7) by a least-mean-square (LMS). This results in magnitudes of $\tau_{max}$ and $s_K$. Then, the probing Thevenin voltage magnitude is assigned to calculate dimensionless coefficients $\alpha$ and $\beta$. The probing Thevenin voltages are determined by consistently increasing the value of the coefficient $\gamma$. Its value is selected from sequentially ascending magnitudes laying in between a relatively narrow range from 0.85–0.87 to

0.97–0.99. The increment of the $\gamma$ coefficient is chosen as per the accuracy requirements, and, as usual, it would not be less than 0.005–0.01. Therefore, the number of probing points remains relatively small, ensuring a short calculation time. Once selected, the coefficient $\gamma$ defines the sequential magnitude of $U_{Th}$ (2) that allows the solution of the combined Equations (5)–(7):

$$\alpha = \frac{3}{2} \frac{U_{Th}^2}{\omega_S \tau_{max} R_2} - \frac{1}{s_K}; \tag{10}$$

$$\sqrt{\alpha^2 + \beta^2} = \frac{1}{s_K} \tag{11}$$

**Table 1.** Results of the AC motor measurement during loading.

| n | 1 | 2 | 3 | ... | N |
|---|---|---|---|-----|---|
| $(\tau_m)_n$, Nm | $\tau_1$ | $\tau_2$ | $\tau_3$ | ... | $\tau_n$ |
| $(\omega_m)_n$, rad/s | $\omega_1$ | $\omega_2$ | $\omega_3$ | ... | $\omega_n$ |
| $(I_{in})_n$, A | $(I_{in})_1$ | $(I_{in})_2$ | $(I_{in})_3$ | ... | $(I_{in})_n$ |
| $(P_{in})_n$, W | $(P_{in})_1$ | $(P_{in})_2$ | $(P_{in})_3$ | ... | $(P_{in})_n$ |
| $(cos\, \varphi)_n$ | $(cos\, \varphi)_1$ | $(cos\, \varphi)_2$ | $(cos\, \varphi)_3$ | ... | $(cos\, \varphi)_n$ |

The p.u. values of $\alpha$ and $\beta$ after the solution are expressed in terms of the variable $R_2$. In Equations (10) and (11), the Thevenin voltage $U_{Th}$ has been defined by the coefficient $\gamma$; the synchronous speed $\omega_S$ is given by the motor template; whereas $\tau_{max}$ and $s_K$ have been assessed previously.

Therefore, assigning an arbitrarily chosen value of $R_2$ in Equation (9) allows the calculation of coefficients $\alpha$ and $\beta$ from Equations (10) and (11). Being assessed coefficients, $\alpha$ and $\beta$ allow the calculation of all circuit parameters. An essential point contributing to an increase in the accuracy and speed of a solution is the narrow range of possible $R_2$ values, which can be significantly limited by the following considerations. The feasible range of resistance $R_2$ is restricted, as the solution of $\alpha$ and $\beta$ must be positive and larger than 1. Thus:

$$\alpha = \frac{3}{2} \frac{U_{Th}^2}{\omega_S \tau_{max} R_2} - \frac{1}{s_K} > 1 \;\Rightarrow\; (R_2)_{max} < \frac{1.5 U_{Th}^2}{\omega_S \tau_{max} \left( \frac{1}{s_K} + 1 \right)}; \tag{12}$$

For $\beta$:

$$\beta^2 = \frac{1}{s_K^2} - \alpha^2 > 1 \;\Rightarrow\; (R_2)_{min} > \frac{1.5 U_{Th}^2}{\omega_S \tau_{max} \left[ \frac{1}{s_K} + \sqrt{\frac{1}{s_K^2} - 1} \right]}; \tag{13}$$

Equations (12) and (13) define the upper and lower limits of $R_2$, between which it provides feasible values of other parameters. The feasible span is relatively narrow, ensuring a relatively small number of probing points for the calculation of equivalent circuit parameters. This circumstance increases the accuracy of the method and the convergence stability of the solution. It is worth noting that for any value of $R_2$ solving Equations (12) and (13), the motor characteristic curve passes exactly through two important points: (0, 0) and ($\tau_{max}$, $s_K$). This, together with the motor characteristic pattern, ensures the closeness of real and theoretical curves between 0–$s_K$ values. However, this is not enough for the determination of all circuit parameters, as the closeness of mechanical curves does not guarantee the coincidence of the theoretical and real motor currents.

Thus, in the next step, the optimal selection of $R_2$ ensures the alignment of the theoretical and real motor currents. It is worth remembering that, once being chosen, $R_2$ provides the basis for the calculation of dimensionless coefficients $\alpha$ and $\beta$ and, therefore, all parameters ($R_1$, $R_2$, $X_1$, $X_2$, and $X_m$) of an equivalent circuit. The optimal $R_2$ magnitude

is chosen from a restricted range of allowable values (12)–(13) by a sequential procedure aiming to obtain a maximal closeness between the theoretical and real motor currents. Different indicators may be chosen for the assessment of the closeness (coincidence) between the theoretical and real currents. The minimal sum of squares of deviations (LMS) between the theoretical curve and experimentally measured current values was selected as an indicator of a coincidence. Obviously, for each arbitrarily chosen $U_{Th}$ the closeness of the experimental current to its theoretical prediction was different.

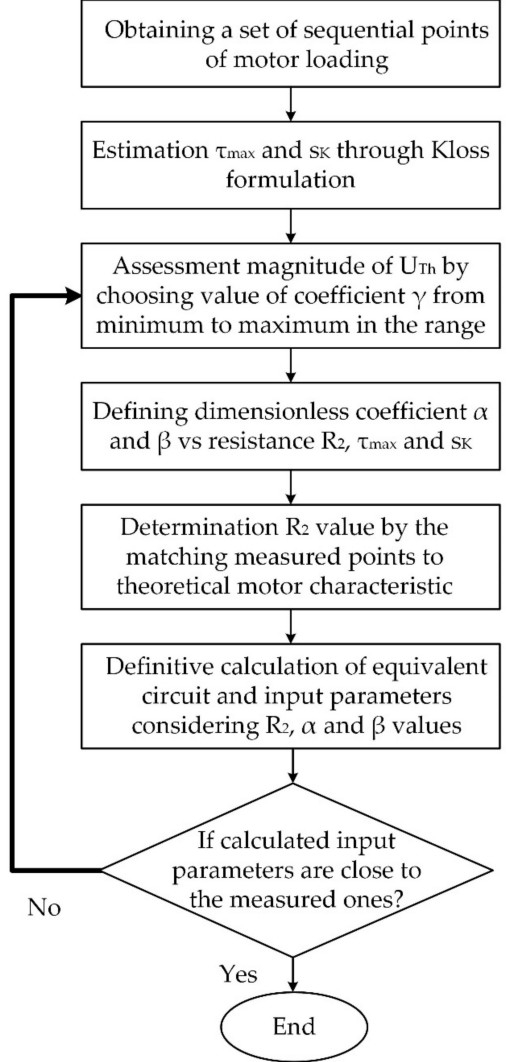

**Figure 3.** Algorithm of the equivalent circuit estimation.

As the optimal $U_{Th}$, the value that ensures the best coincidence between theoretical and real currents should be chosen. It is worth noting that the Thevenin voltage is defined by the value of the coefficient $\gamma$ (2). Therefore, selecting the coefficient $\gamma$ arbitrarily provides the best $R_2$ magnitudes and thus defines all equivalent motor parameters. The result of assigning $\gamma$ sequentially (within its feasible range 0.9–0.99) is the set of the criterion (LMS) values defining the closeness between theoretical and real currents. The global minimum of this criterion determines the global optimal $R_2$ value and hence all equivalent parameters of the equivalent circuit.

Owing to the procedure mentioned above, the algorithm of equivalent circuit parameters finding has been established (Figure 3).

The algorithm begins with the determination of $\tau_{max}$ and $s_K$ by the Kloss formalization. Then, for each magnitude of coefficient $\gamma$, the optimal $R_2$ value is chosen. In the end, the

optimal pair of $\gamma$ ($U_{Th}$) and $R_2$ is determined, and all equivalent circuit parameters are calculated following the algorithm's sequence.

The estimation of $\tau_{max}$ and $s_K$ due to the obtained measured loading points (see Table 1) of a motor characteristic should be evaluated by the Kloss expression with the use of the LMS approach. The following is represented in Table 1. The first row consists of a sequential number of tests—n. The second row includes the measured motor torque $(\tau_m)_n$ in each test. The third row includes the measured motor velocities $(\omega_m)_n$. The fourth shows the motor currents $(I_{in})_n$. The fifth shows the input power $(P_{in})_n$. The sixth and last row shows the measured power factors $(cos\ \varphi)_n$.

This task of $\tau_{max}$ and $s_K$ is not trivial, as the Kloss equation is not linear, and the application of LMS requires rigorous mathematical development. Therefore, this procedure is discussed in detail in the following section.

### 2.3. Matching a Kloss Formula to the Measured Data of Motor Loading

The LMS protocol assumes the minimum of summarized squared deviations between theoretical and measured data (Equation (7)). Therefore, the assessment of parameters $\tau_{max}$ and $s_K$ is ensured by the minimum of the sum that is stated for the set of experimental data (Table 1) by Equation (14). This procedure of finding $\tau_{max}$ and $s_K$ should be the first priority.

$$Z = \sum_{i=1}^{N}\left(\tau_i - \frac{2\tau_{max}}{\frac{s_i}{s_K} + \frac{s_K}{s_i}}\right)^2 \Rightarrow MIN \tag{14}$$

The solution of Equation (14), requiring the minimization of the sum of squares of deviations, is obtained by equalizing the partial derivatives of Z for $\tau_{max}$ and $s_K$ to zero. The result of the differentiation is as follows:

$$\frac{dZ}{\tau_{max}} = -\sum_{i=1}^{N}\left(\tau_i - \frac{2\tau_{max}}{\frac{s_i}{s_K} + \frac{s_K}{s_i}}\right)\left(\frac{1}{\frac{s_i}{s_K} + \frac{s_K}{s_i}}\right) = 0$$

$$\frac{dZ}{s_K} = -\sum_{i=1}^{N}\left(\tau_i - \frac{2\tau_{max}}{\frac{s_i}{s_K} + \frac{s_K}{s_i}}\right)\frac{2\tau_{max}\left(\frac{1}{s_i} - \frac{s_i}{s_K^2}\right)}{\left(\frac{s_i}{s_K} + \frac{s_K}{s_i}\right)^2} = 0 \tag{15}$$

The transformation of these expressions causes the algebraic fifth-order equation to be impossible for an analytical solution. However, the absolute minimum of Equation (14) in terms of $\tau_{max}$ and $s_K$ exists, which can be demonstrated, for example, by Figure 4. An analysis of Equation (14) shows the presence of one global minimum.

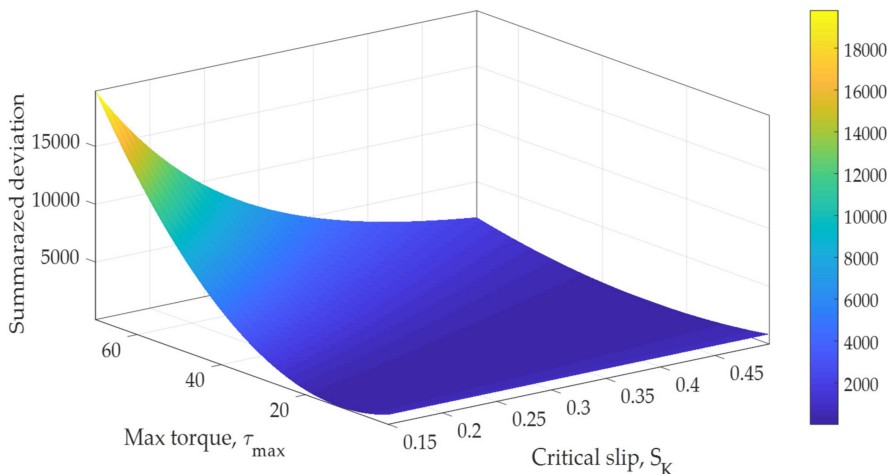

**Figure 4.** 3-D graph of the summarized error between real AC motor data and their approximation by the Kloss formula with different values of the maximum torque $\tau_{max}$ and critical slip $s_K$.

Considering only one minimum value of the maximum motor torque $\tau_{max}$ and critical slip $s_K$, the minimum of Equation (14) can be obtained by the numerical approach relatively simply. The developed scanning routine (MATLAB) searches $\tau_{max}$ and $s_K$ values within their feasible ranges and can find these values with any required precision.

## 3. Results

### 3.1. Validation of the Submitted Approach

The validation of the submitted method was carried out in an electrical machine laboratory. The testbench for the experiments and measurements is shown in Figure 5a,b.

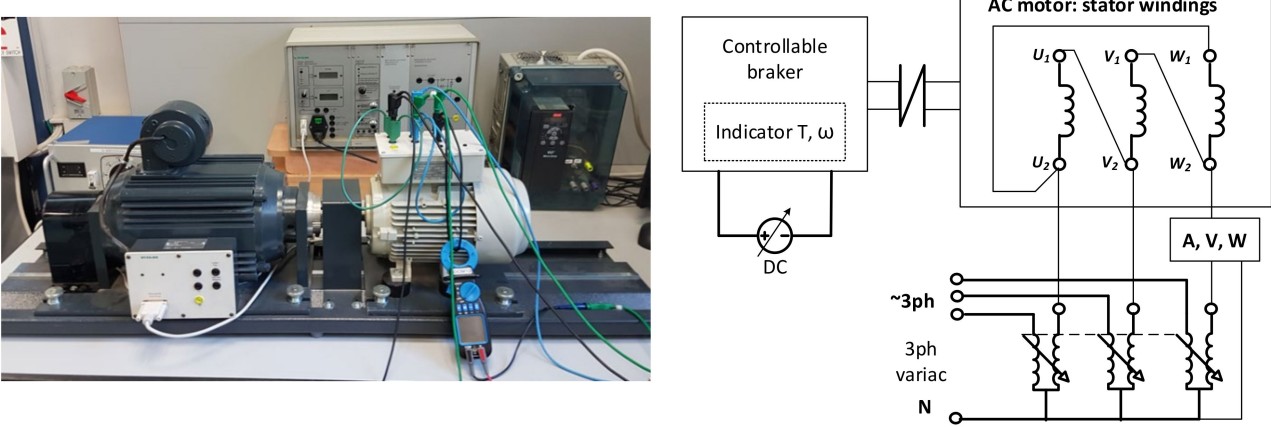

(**a**) Experimental testbench          (**b**) The scheme of the experimental setup

**Figure 5.** The testbench (**a**) for the experiments on AC motor characterization and the electrical scheme of the setup (**b**).

The testbench included a 3ph AC motor with the number of poles $P = 4$ (synchronous speed $n_{synch} = 1500$ rpm), a stator winding connection delta (D), $(U_{ph})_{nom} = 400$V, $f = 50$ Hz, $P_{nom} = 1000$ W, and $I_{nom} = 1.8$ A. The AC motor was fed by a 3ph variac for the control motor supply. The motor was attached to a controllable braking unit capable of applying a load torque in a wide range (0–20 Nm) and equipped with torque and rotation speed indicators. VoltAmperWatt meters were connected in the scheme for the measurement and registration of the input current, voltage, active power, and *cos (φ)* during experiments. The accuracy of the measurements of the motor torque was less than ±3%, the rotation velocity was less than ±0.5%, and other electrical parameters were not less than 0.4%. During experiments, the AC motor was loaded increasingly from the minimum up to the maximum allowable torque, and the measured parameters were recorded.

### 3.2. Representation of the Approach

To apply the submitted approach, the loading motor test with the measurement of input–output parameters ($U_{ph}$, $I_{ph}$, $P_{in}$, $cos\ \varphi$, $n_m$, and $\tau_m$) should be carried out first. The maximum torque $\tau_{max}$ and critical slip $s_K$ values should be evaluated in the next step based on a measured array of $n_m$ and $\tau_m$. Finding parameters $\tau_{max}$ and the slip $s_K$ ensures in principle the closeness of the theoretical output motor characteristic ($n_m = \Phi(\tau_m)$) to measured points and the is basis for the estimation of all circuit parameters. It is done with the numerical approach that is facilitated by the theoretical analysis provided by the Thevenin circuit. The variable $\gamma$ is chosen arbitrarily from within its practicably valid range. Immediately after the selection of coefficient $\gamma$, the feasible limits of a resistance $R_2$ are determined, between which additional probing and scanning is carried out. The definite set of other circuit parameters for each $R_2$ value is then obtained as described in the methodology section. It is worth mentioning that the set of equivalent parameters determines the theoretical motor current function vs. the motor torque that can be com-

pared with the measurement data with the assessing of the cost function value for this current curve. Arbitrary scanning of the 2-D space of coefficient $\gamma$ and $R_2$ finds their most applicable magnitudes, providing the best matching of a theoretical motor current to its measured values. The abovementioned results of the laboratory measurements of a given AC motor are represented in Table 2.

**Table 2.** Results of the experimental AC motor testing.

| $V_{ph}$, V | $n_m$, rpm | $I_{ph}$, A | $P_{ph\_measured}$, W | $\cos\varphi$ | $\tau_m$, Nm | $P_{3ph}$, W | $P_{mech}$, W | $\eta$ |
|---|---|---|---|---|---|---|---|---|
| 400 | 1470 | 1.42 | 84 | 0.15 | 0.33 | 252 | 51.3 | 0.204 |
| 400 | 1469 | 1.4 | 93 | 0.16 | 0.51 | 279 | 78.2 | 0.280 |
| 400 | 1466 | 1.39 | 118 | 0.21 | 1.10 | 354 | 168.9 | 0.477 |
| 400 | 1455 | 1.37 | 159 | 0.28 | 2.08 | 477 | 317.4 | 0.665 |
| 400 | 1453 | 1.36 | 195 | 0.36 | 2.95 | 585 | 448.9 | 0.767 |
| 400 | 1444 | 1.38 | 253 | 0.46 | 4.21 | 759 | 636.3 | 0.838 |
| 400 | 1430 | 1.435 | 331 | 0.58 | 5.83 | 993 | 873.5 | 0.880 |
| 400 | 1416 | 1.53 | 410 | 0.67 | 7.42 | 1230 | 1099.7 | 0.894 |
| 400 | 1384 | 1.82 | 578 | 0.8 | 10.12 | 1734 | 1466.2 | 0.846 |
| 400 | 1351 | 2.15 | 730 | 0.85 | 11.92 | 2190 | 1685.9 | 0.770 |
| 400 | 1345 | 2.16 | 740 | 0.86 | 11.97 | 2220 | 1685.4 | 0.759 |
| 400 | 1300 | 2.53 | 900 | 0.89 | 13.13 | 2700 | 1787.9 | 0.662 |
| 400 | 1277 | 2.82 | 1015 | 0.9 | 13.99 | 3045 | 1845.4 | 0.606 |

Following the algorithm, the parameters of the maximum motor torque $\tau_{max}$ and critical slip $s_K$ were first estimated by a special MATLAB routine, providing the matching of the Kloss formula to the operating points. These parameters, based on the results of Table 2, were obtained as $\tau_{max}$ = 15.9 Nm and $s_K$ = 0.254 p.u. accordingly. The output characteristic of this AC motor built with the Kloss formula and operating points of a motor is shown in Figure 6. A relatively good coincidence between the theoretical characteristics and real measurements can be observed: the discrepancy was less than 6%. Despite this discrepancy, which was relatively small, this method is good enough for most engineering applications.

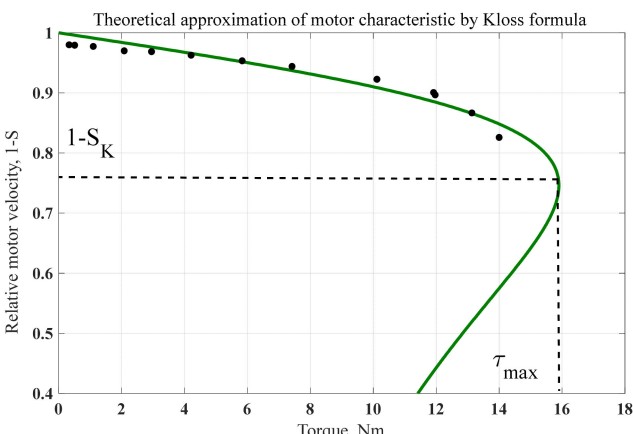

**Figure 6.** The AC motor characteristic described by the Kloss expression compared with the measured points.

Further, following the algorithm flowchart, lower and upper limits for the $R_2$ resistance are calculated considering the obtained $\tau_{max}$ and $s_K$ for different $U_{Th}$ manifested in terms of the coefficient $\gamma$ (Equation (2)). The results of the calculations are represented in Table 3.

**Table 3.** Upper and lower limits of the resistance $R_2$ for different $U_{th}$ voltages.

| $\gamma$, p.u. | 0.99 | 0.98 | 0.97 | 0.96 | 0.95 | 0.94 | 0.93 | 0.92 | 0.91 | 0.90 | 0.89 |
|---|---|---|---|---|---|---|---|---|---|---|---|
| $U_{th}$, V | 396 | 392 | 388 | 384 | 380 | 376 | 372 | 368 | 364 | 360 | 356 |
| $(R_2)_{min}$, $\Omega$ | 12.15 | 11.91 | 11.67 | 11.43 | 11.19 | 10.95 | 10.72 | 10.49 | 10.27 | 10.04 | 9.82 |
| $(R_2)_{max}$, $\Omega$ | 19.06 | 18.68 | 18.30 | 17.92 | 17.55 | 17.19 | 16.82 | 16.46 | 16.11 | 15.75 | 15.41 |

Later, for each coefficient $\gamma$, the optimal $R_2$ magnitude was chosen by a searching numerical procedure inside a permissible range that is determined by Equations (10) and (11). This optimal magnitude provides the minimal discrepancy between the theoretical and real motor currents. During the search procedure for the changing $\gamma$, the series of quantitative indicators formulated by the LMS approach and the definition of the discrepancy between the theoretical and real currents can be obtained. Therefore, as per the cost function for the optimal selection of the dimensionless parameter $\gamma$ and reduced resistance $R_2$ of the rotor circuit, the sum of squares of deviations of theoretical and real motor current was chosen. The smallest of these sums determine the optimal magnitude of a coefficient $\gamma$ and resistance $R2$. The selected optimal parameters of $\gamma$ and $R_2$ allow the assessment of the most appropriate value $U_{th}$ and then all the missing additional parameters of the equivalent circuit: $R_1$, $X_1$, $X_2$, and $X_m$. The determination of equivalent parameters that provide the minimum of the discrepancy between the theoretical and real currents ensures the additional exactness of the approach presented in this paper.

Special MATLAB software was developed for finding the equivalent circuit parameters based on the proposed algorithm. The results of the software application were as follows: $R_1 = 20.35\ \Omega$; $R_2 = 15.92\ \Omega$; $X_1 = X_2 = 30.18\ \Omega$; and $X_m = 335.3\ \Omega$. An analysis of the obtained results demonstrates a coincidence of output characteristics of 98–99%. The average square root deviation for a motor theoretical current is about 11%. The graph of the measured phase motor current and its theoretical assessment is represented in graph Figure 7.

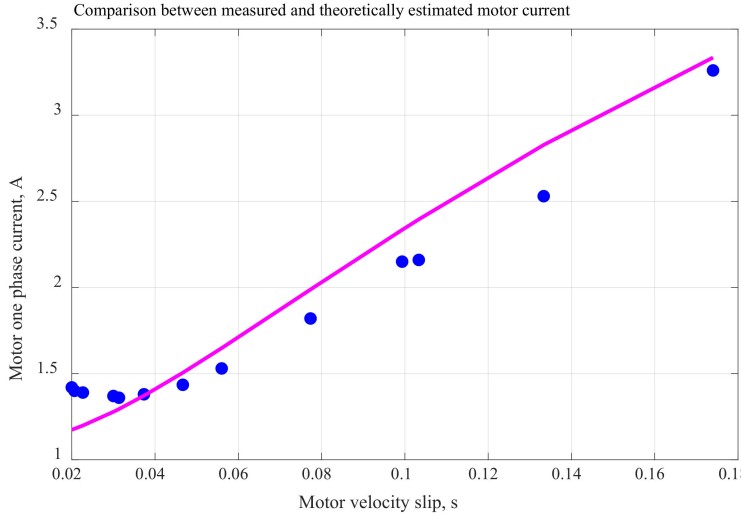

**Figure 7.** Measured phase motor current (blue dots) and its theoretical assessment.

The deviation between the theoretical and measured currents was not significant. Therefore, our method of finding the parameters of the AC motor equivalent circuit can be applied effectively to various applications of electrical machines and drives.

## 4. Discussion

The task of determining an equivalent AC motor circuit is not trivial and represents a significant mathematical challenge due to the requirements for obtaining an accurate and stable solution. The relatively accurate and stable solution of an equivalent T-circuit of an AC induction motor can be reached with the application of the analytical searching procedure, leading to the decisive algorithm is presented in the article. The first part of the algorithm is based on the use of the Thevenin theorem and Kloss formula. They represent the analytical fragment of the algorithm that can provide some of the parameters. The additional part of the algorithm is based on the numerical search procedure providing other missing equivalent parameters.

The algorithm was implemented by specially developed MATLAB software. It is based on the experimentally obtained data: the rotation velocity, torque, and current. The method described in the article suggests a relatively simple approach to finding the parameters of the T-equivalent circuit of an AC induction motor. The application of the submitted MATLAB software significantly improves the accuracy of the assessment and increases the solution's stability. The relative error of output motor characteristics was 11%. The error in the motor current estimation remained between 0 and 11%.

This method can be carried out using the standard equipment of an electrical machine laboratory with the use of MATLAB programming software. The method in this article is particularly relevant to electrical machines and electrical drives courses and other engineering applications, as it can provide high-quality results.

## 5. Conclusions

The solution to the assessment of T-equivalent circuit parameters of an induction motor can be effectively based on the successfully found combination of analytical and numerical approaches. Such a finding represents the significant novelty of the method as ensuring a fast convergence and high accuracy of the parameters' estimation.

As per the analytical solution, the Thevenin theorem with a Kloss formulation in dimensionless terms was applied to provide a rigorous definition of the output circuit parameters. The analytical part of the solution diminishes the probing space of a numerical search for supplementary circuit parameters and increases the stability of the finding

The main idea of the method can be applied further to the more complicated equivalent circuit of AC induction motors.

**Author Contributions:** Conceptualization, M.A.; methodology, M.A. and E.L.; software, M.A.; validation, E.L.; formal analysis; investigation, E.L.; writing—original draft preparation, M.A.; writing—review and editing, M.A.; visualization, E.L. All authors have read and agreed to the published version of the manuscript.

**Funding:** This research received no external funding.

**Informed Consent Statement:** Not applicable.

**Data Availability Statement:** Not applicable.

**Acknowledgments:** The authors wish to acknowledge the technical support of engineer Arieh Shochat in carrying out laboratory experiments and measurements and Judy Frank for the help in language editing and improvements.

**Conflicts of Interest:** The authors declare no conflict of interest.

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
