# Peer review of "Estimation of the Equivalent Circuit Parameters of Induction Motors by Laboratory Test"

_machines, doi:10.3390/machines9120340_

Round 1

Reviewer 1 Report

This paper proposes an original approach of AC motor testing based on the data of motor loading, the Kloss equation, and the Thevenin circuit. The simplified experimental technique ensures accurate testing results.

1. The main remark regarding the presented manuscript is some lack of clarity in the scientific novelty. It should be emphasized in more details, especially in Abstract and Conclusions. There are some additional remarks as follow.

2. The authors have mentioned about the optimal selection of R2, however the cost fucntion and optimization approach should be discussed and explained in more detail.

3. The equations numeration should be corrected. Currently equation (2) is followed by equation (5) and also (5.1) and (5.2) are marked. The same with (7.1-7.2).

4. The quality of figure 3 should be improved.

5. The results in Table 1 should be presented in more reader-freindly way. It was also written "Results of AC motor measurement during loading"m but variables instead of values make readers confusing.

Author Response

Reviewer N1

Comments and Suggestions for Authors

This paper proposes an original approach of AC motor testing based on the data of motor loading, the Kloss equation, and the Thevenin circuit. The simplified experimental technique ensures accurate testing results.

First, thank you for your kind estimation of our work.

  1. The main remark regarding the presented manuscript is some lack of clarity in the scientific novelty. It should be emphasized in more details, especially in Abstract and Conclusions. There are some additional remarks as follow.

Thank you for your useful comment. We included the required explanation in the ABSTRACT as well as added the section Conclusions. All changes are signed in a blue color inside a text.

Abstract: Determination of equivalent circuit parameters for AC induction motors represents an important task in an electrical machine laboratory. Frequently used open-circuit and short current tests answer these requirements. However, the results have low accuracy. It becomes especially obvious when the equivalent circuit is applied for the motor current and power predicting. The main obstacles for this circumstance lie in the difficulty of providing a pristine open-circuit test, the lack of which causes errors in parameter estimation. A much more accurate approach can be carried out with a test including several output points with measurements of motor torque, velocity, current, and power magnitudes. Nevertheless, a relatively simple and accurate method to ensure determining parameters for such tests does not exist. This article tries to provide such a method by an approach based on Kloss's simplified equation and Thevenin theorem. The significant novelty of the method is specially selected synergetic interaction between analytical and numerical approaches giving a relatively simple algorithm with good accuracy and convergence of parameters estimation.

And then:

  1. Conclusions

The solution for the assessment of T-equivalent circuit parameters of induction motor can be effectively based on the successfully found combination of analytical and numerical approaches. Such a finding represents the significant novelty of the method as ensuring fast convergence and high accuracy of parameters estimation.

As per the analytical solution, the Thevenin theorem with a Kloss formulation in the dimensionless terms was applied to provide the rigorous definition of the output circuit parameters. The analytical part of the solution diminishes the probing space of a numerical search for supplementary circuit parameters and increases the stability of the finding   

The main idea of the method can be further applied for a more complicated equivalent circuit of AC induction motors.

  1. The authors have mentioned about the optimal selection of R2, however the cost fucntion and optimization approach should be discussed and explained in more detail.

Thank you for this comment. We considered this remark and changed the text as:

L.300-305:

Therefore, as per the cost function for the optimal selection of dimensionless parameter γ and reduced resistance R2 of the rotor circuit was chosen the sum of squares of deviations of theoretical and real motor current. The smallest of these sums determines the optimal magnitude of a coefficient γ and resistance R2. The selected optimal parameters of γ and R2 allow the assessment of the most appropriate value Uth and then all the missing additional parameters of the equivalent circuit: R1, X1, X2, and Xm. The determination of equivalent parameters that provide the minimum of theoretical and real currents discrepancy ensures the additional exactness of the approach presented in this paper.

  1. The equations numeration should be corrected. Currently equation (2) is followed by equation (5) and also (5.1) and (5.2) are marked. The same with (7.1-7.2).

Thank you for this comment. We regret this technical flaw. The expressions numeration was changed in the text as per the request.

  1. The quality of figure 3 should be improved.

Thank you for this comment. The quality of figure 3 was improved.

  1. The results in Table 1 should be presented in more reader-freindly way. It was also written "Results of AC motor measurement during loading"m but variables instead of values make readers confusing.

Thank you for this comment.

For the clarification of the parameters in Table 1 we added in the text a detailed description of all included parameters. (L.324-333).

Thank you once again for your fruitful and important work!

Reviewer 2 Report

Dear Authors,

The paper deals with an interesting topic for all the researchers interested in the behavior of the IM. I have few observations:

  1. The term AC in the title is strange as the IM is an AC electric motor.
  2. I think that a short presentation of the sections of the paper is necessary.
  3. In my opinion, you should present clearly the starting data necessary to apply this method.
  4. I consider that you should present the advantages of the proposed method. You present experimental data at the beginning of subsection 3.2. The method is quite complicated and the presentation of its advantages over other methods is important.
  5. Please correct the sentence on line 34 of the paper (The first one is includes …)

As a general remark, I consider that you need to present more clearly the method to evidence better its utility.

Best regards

Author Response

Reviewer N2

Comments and Suggestions for Authors

Dear Authors,

The paper deals with an interesting topic for all the researchers interested in the behavior of the IM.

First, thank you for this kind estimation of our work.

I have few observations:

1. The term AC in the title is strange as the IM is an AC electric motor.

Thank you for this comment. We exclude the word “AC” from a title.

2. I think that a short presentation of the sections of the paper is necessary.

Thank you for this comment.

Brief representation of the sections in the article is done in L.79-91.

3. In my opinion, you should present clearly the starting data necessary to apply this method.

Thank you for this comment.

The text was added with the explanation of the starting data necessary to apply this method – L.259-275.

4. I consider that you should present the advantages of the proposed method. You present experimental data at the beginning of subsection 3.2. The method is quite complicated and the presentation of its advantages over other methods is important.

Thank you for this comment.

We expect that the including Conclusions section can facilitate the advantages of the method. Moreover, the additional text (L.:259-275) which was included as per your previous comment serves the same aim to improve the explanation of the applicability and advantages of the approach.

5. Please correct the sentence on line 34 of the paper (The first one is includes …)

Thank you for this comment.

The change was done in the text.

As a general remark, I consider that you need to present more clearly the method to evidence better its utility.

Thank you for this comment.

We hope that the additionally included parts in the text which was explained in the previous answers to your useful remarks can clarify the evidence of the approach utility.

 Best regards

Thank you once again for your fruitful and important work!

Reviewer 3 Report

The paper deals with the estimation of the equivalent circuit parameters of an induction motor. 
The description of the proposed algorithm is quite clear. Experimental results confirm the validity of the proposed method.
Some minor corrections are necessary:
(a) the numbering of the formulas is not correct: after  (2), we go to number (5) 
b) the expression of Uth is not complete

Author Response

Reviewer N3

Comments and Suggestions for Authors

The paper deals with the estimation of the equivalent circuit parameters of an induction motor. 
The description of the proposed algorithm is quite clear. Experimental results confirm the validity of the proposed method.

Thank you for this kind estimation of our work!

Some minor corrections are necessary:
(a) the numbering of the formulas is not correct: after  (2), we go to number (5) 

Thank you for this remark. The correction of the formulas notation was done.

  1. b) the expression of Uth is not complete

Thank you for this remark.

The formula for UTh was represented in expression (2). However, for clarification, we divided (2) into three separated representations for UTh, RTh, and XTh in (2.1), (2.2), and (2.3) accordingly.

Thank you once again for your fruitful and important work!

Round 2

Reviewer 1 Report

The manuscript has been improved and modified. All remarks were addressed. The paper could be accepted for publication.

Reviewer 2 Report

Dear Authors,

Thank you for answering in detail to all my remarks.

Best regards